# Modeling of Polymer Composite Materials Chaotically Reinforced with Spherical and Cylindrical Inclusions

**DOI:** 10.3390/polym14102087

**Published:** 2022-05-20

**Authors:** Kristina Berladir, Dmytro Zhyhylii, Oksana Gaponova, Jan Krmela, Vladimíra Krmelová, Artem Artyukhov

**Affiliations:** 1Department of Applied Materials Science and Technology of Constructional Materials, Sumy State University, 2, Rymskogo-Korsakova St., 40007 Sumy, Ukraine; gaponova@pmtkm.sumdu.edu.ua; 2Department of Computational Mechanics Named after Volodymyr Martsynkovskyy, Sumy State University, 2, Rymskogo-Korsakova St., 40007 Sumy, Ukraine; d.zhigiliy@omim.sumdu.edu.ua; 3Faculty of Mechanical Engineering, J. E. Purkyně University in Ustí nad Labem, Pasteurova 1, 400 96 Ustí nad Labem, Czech Republic; jan.krmela@ujep.cz; 4Faculty of Industrial Technologies in Púchov, Alexander Dubček University of Trenčín, I. Krasku 491/30, 02001 Púchov, Slovakia; vladimira.krmelova@tnuni.sk; 5Academic and Research Institute of Business, Economics and Management, Sumy State University, 2, Rymskogo-Korsakova St., 40007 Sumy, Ukraine; a.artyukhov@pohnp.sumdu.edu.ua

**Keywords:** PCMs, polytetrafluoroethylene, carbon fiber, coke, energy efficiency, modeling, strength, solid model, finite element model

## Abstract

The technical and economic efficiency of new PCMs depends on the ability to predict their performance. The problem of predicting the properties of PCMs can be solved by computer simulation by the finite element method. In this work, an experimental determination of the physical and mechanical properties of PTFE PCMs depending on the concentration of fibrous and dispersed filler was carried out. A finite element model in ANSYS APDL was built to simulate the strength and load-bearing capacity of the material with the analysis of damage accumulation. Verification of the developed computer model to predict the mechanical properties of composite materials was performed by comparing the results obtained during field and model experiments. It was found that the finite element model predicts the strength of chaotically reinforced spherical inclusions of composite materials. This is due to the smoothness of the filler surfaces and the lack of filler dissection in the model. Instead, the prediction of the strength of a finite element model of chaotically reinforced cylindrical inclusions of composite materials requires additional analysis. The matrix and the fibrous filler obviously have stress concentrators and are both subject to the difficulties of creating a reliable structural model.

## 1. Introduction

The use of composite materials based on polymers is an important factor in improving the efficiency and successful development of leading industries [1]. At the same time, the rapid development and improvement of composite materials, as well as the constant need for modern production in them, urgently require the creation of competitive materials with predictable properties [2,3] and the development of high-quality polymer composites with specified technical characteristics [4,5,6]. 

Polymer composite materials (PCMs) based on amorphous-crystalline linear polymers such as polytetrafluoroethylene (PTFE) [7,8] are currently widely used as structural materials, including for tribotechnical purposes [9]. They have unique antifriction properties, high chemical inertness, and heat and cold resistance [10]. The advantage of these materials is the higher values of their mechanical properties compared to amorphous polymers [11]. However, low wear resistance and insufficient mechanical properties limit the possibility of their use in unfilled form [12]. For PTFE, there is a possibility for the effective purposeful regulation of operational characteristics by filling [13,14,15]. It has been practically established [16,17,18] that in order to maintain at least 60–70% strength and 50% relative elongation of pure PTFE, the volume fraction of the introduced fillers should not exceed 20%.

The main requirement that must be met by the filler for PTFE is the ability to withstand heat to a temperature of 663 K, at which the sintering of PTFE products is performed [19]. Using fillers different in chemical nature and condition, for the same polymer matrix, can lead to radically different properties of materials. Fibrous fillers reduce the range of crack formation in the volume of the composite [20,21], and dispersed ones increase the energy threshold of crack formation (fracture) [22,23].

The use of the fillers is not limited to polymer matrices. Recently, promising research results on their addition to polymer concrete [24] and geopolymer concrete [25] were obtained. Yu P. et al. [24] investigated the effects of the content of crumb rubber, chopped glass fibers, and polypropylene fibers when implemented in an analytical hierarchy process to determine the most suitable polymer mix for a railway sleeper application. The paper [25] considered the impact of four types of short fibers (twisted polypropylene fibers, straight polypropylene fibers, short glass fibers, and steel fibers) in different dosages on the optimum strength of geopolymer mortar.

Numerous studies in the field of PTFE-based polymer composite materials have convincingly proven the legitimacy of the use of carbon fiber (CF) fillers of various nature with a wide range of special properties [26,27,28,29,30]. Currently, antifriction carbon fiber-reinforced PCMs based on PTFE have been developed and successfully used in friction units of various equipment, including compressor equipment [31,32,33]. It was found that their optimal content in a PTFE composite varies in a wide range (5–20) wt.% depending on the brand of filler [34,35].

Metal powders, graphite, coke, etc., increase the thermal conductivity of the composition, which has a positive effect on the performance of friction units [36,37,38]. It was found that the minimum wear during dry friction of a PTFE composite is observed at (20–30) wt.% of the content of most dispersed fillers [39,40]. PTFE with the addition of 20 wt.% of coke in comparison with unfilled has 600 times greater wear resistance and a third greater rigidity [18,41].

Composite materials based on PTFE have a potentially wide range of performance properties, which is provided using various technological methods of obtaining fillers and the composition as a whole [42,43,44]. Due to the high viscosity of the melt, thermoplastic PTFE and compositions based on it cannot be processed into a product by worm extrusion or injection molding [45]. Therefore, the methods of processing compositions based on PTFE into composite are based on a two-stage process: obtaining the workpiece by pressing and subsequent heat treatment of the workpiece. In this way, the blanks of the simplest forms are obtained, from which the finished products are made by machining on metal-cutting machines.

PTFE provides good self-lubricating properties of the composite, but is characterized by low surface energy [46,47]. This makes it difficult to obtain a strong adhesive bond between the filler and the matrix, which dramatically reduces the physical and mechanical properties of composites when fillers are introduced into their composition [48]. The only way to create a strong adhesive bond is to structurally modify the surface of the polymer and the filler.

PTFE should be modified by mechanical activation, i.e., changing its physical and mechanical properties without changing the chemical composition of the polymer and its molecular weight, i.e., only changing the supramolecular structure of the polymer. Mechanical activation is used to change the reactivity of solids, which means accelerating or increasing the efficiency of chemical or physical processes [49]. This technology is widely used in industry for the activation of small molecules (e.g., ball grinding) due to the low energy and metal consumption of equipment, simplicity, and safety of the process [50]. It was demonstrated that composites based on PTFE with mechanically activated components are significantly superior in terms of the strength and durability of materials of similar composition obtained by traditional technology [14,32,35,42,44,51,52]. However, mechanochemical technologies for the modification of PFTE have not yet found industrial application, although the results of laboratory studies are encouraging in terms of increasing the strength of mechanoactivated filled PTFE composites. This indirectly indicates an increased level of adhesion at the PTFE–filler interface. In this work, as a preliminary operation for the preparation of the polymer matrix and fillers before mixing, mechanical activation was used according to the developed modes [52].

Issues of predicting the properties of PCMs occupy a leading place in materials science, because viscoelastic media can significantly affect the physical and mechanical properties of PCMs [53]. Therefore, the technical and economic efficiency of new materials depends on the ability to predict their performance. 

The problem of predicting the properties of PCMs can be solved both by developing new and effective mathematical and numerical research methods [54] and with the level of understanding of the physicochemical processes that determine the mechanical properties of these complex materials [55]. Methods for calculating composite materials are based on solving the equations of solid medium mechanics. It can be done using analytical approaches [56], as well as using finite element methods, boundary elements, finite differences, finite volumes, and other numerical methods applicable to specific tasks [57].

The mechanics of PCM hardening use the mechanism of stress transfer between the softer matrix and the stiffer filler during loading [58]. Stress transfer occurs at the polymer–filler interface. Therefore, the structure and properties of interfacial surfaces play an important role in the mechanical properties of PCMs. Due to the complexity of load transfer, solving the problem of predicting the properties of PCMs by analytical methods can be problematic. The main numerical method for solving problems of composite mechanics is currently the finite element method (FEM) [59].

Thus, research to improve the methods of modeling the behavior of structures made of composite materials, considering their structure at different levels of the organization, will optimize their resistance to destruction, strength, fluidity, wear resistance, and other parameters. In addition, computer models are easier and more convenient to study; they allow computational experiments, the actual formulation of which is difficult or can give unpredictable results. The logic and formalization of computer models allow us to identify the main factors that determine the properties of the studied objects, and to study the response of the physical system to changes in its parameters and initial conditions.

There is already a great deal of well-represented work on the modeling and analysis of multilayer composite structures such as fiber-reinforced plastic, multi-layer plates, sandwich panels, etc. The analysis of multilayer composite shells is entirely based on the three-dimensional theory of elasticity or the theory of the equivalent layer. Thus, it can be realized using the tools and functionality of the ANSYS Composite PrepPost module, for example, or in other special software. Instead, there are a limited number of articles on the modeling of composite materials, in which there is no strict procedure for filling with fillers of different shapes and sizes. Such studies [60,61] focus on the computational methodology for generating microstructure models of random composite inclusions, without considering the real structure and properties of the matrix material.

Thus, there is an urgent need for research on the modeling of composite materials with matrices of various polymers and randomly filled with fibrous (cylindrical) and dispersed (spherical) fillers of different nature.

The novelty of this study is that a finite element model was obtained for the first time, which well predicts the strength of materials chaotically filled with spherical inclusions of polytetrafluoroethylene. A finite element model for predicting the strength of polytetrafluoroethylene, which is chaotically reinforced with cylindrical inclusions, requires additional analysis. It will serve as a basis for further improvement.

The aim of the article is to develop an adequate mathematical model for predicting the physical and mechanical properties of PCMs based on PTFE depending on the type of filler using the software package ANSYS and its subsequent verification with experimental data. The following objectives have been formulated to achieve this goal. Firstly, tests to determine the concentration dependence of the physical and mechanical properties of the developed PTFE composites with fibrous and dispersed filler should be performed. Secondly, the solid models of the composite material with spherical and cylindrical inclusions based on the justification of selecting variables should be developed. Finally, a finite element model of the composite material should be constructed and verified with experimental data.

## 2. Materials and Methods

### 2.1. Materials

For the experimental research, as a polymer matrix, we used industrial PTFE for the manufacture of general-purpose products and compositions. 

We studied carbon fibers grade UTM-8-1s, made of hydrated cellulose fabric and obtained by chemical treatment in an aqueous solution of fire retardants Na_2_B_4_O_7_·10H_2_O + (NH_4_)_2_HPO_4_ and annealed at a temperature of (723 ± 20) K in a natural gas medium. Fragments of CF were obtained from UTM-8-1s fabric by cutting and grinding in a hammer crusher KDU-2.0 (3000 rpm) and a mixer MRP-1M (7000 rpm) to an average fiber size (100–150) μm. Obtained fragments of CF were used as fibrous filler.

The finely dispersed casting coal coke brand KL-1, which is a black powder, was used as a dispersed filler.

The initial data on the properties of the experimental materials are given in Table 1.

### 2.2. Technology of Obtaining of PCMs

PTFE powder was prepared by mechanical activation in the dry state in a high-speed mixer in the following mode: the number of revolutions of the working bodies of the mill was *n* = 9000 rpm during the optimal experimentally determined time interval (τ = 5 min).

The experimental mixer is constructed based on a mill for the grinding of dry vegetable samples. Modernization of the mixer consisted in the replacement of the standard asynchronous AC motor with a collector DC motor with electronic control, which allowed us to smoothly change the speed of the working body of the mixer from 0 to 9000 rpm. The control unit of the mixer motor is equipped with a voltmeter and an ammeter to determine the power consumption of the motor during the mixing process. The design of the experimental stand allowed us to quickly replace the mixing chamber of different designs.

Mechanical activation of fillers was carried out in the dry state in a high-speed mixer at the number of revolutions of the working bodies of the mill *n* = 7000 rpm during 5 min for dispersed fillers and 9 min for fibrous.

Mixing of the ingredients of the composition was carried out according to a two-stage scheme: at the first stage of the technological process, we prepared a mixture with a ratio of PTFE:CF/coke = 1:1, which was subjected to intensive mechanical activation in a high-speed mixer and then mixed with the prescribed amount of PTFE (1:4).

Materials for testing were obtained by powder metallurgy technology (Figure 1): cold pressing on the hydraulic press MS-500 (pressing pressure P_pr_ = (50.0–70.0) MPa) with the subsequent sintering of tableted preparations. Sintering of blanks consisted of heating to a temperature of (633–653) K, holding at this temperature (for 1 h per 1 mm of thickness), and rapid cooling of blanks in the temperature range from 600 to 623 K.

After pressing and sintering, the material was kept at room temperature for 15 days, and then, for 24 h, they were conditioned at relative humidity (65 ± 2)%.

### 2.3. Methods for Determining the Properties of PCMs

The method of studying the properties of the composite included the determination of the density ρ (kg/m^3^), breaking strength σ_b_ (MPa), relative elongation at break δ (%), and wear intensity I × 10^−6^ (mm^3^/N·m). Indicators of tensile strength, elongation at break, and wear intensity were the main ones, as potential consumers of tribotechnical composites focus on them. The wear intensity of composite products is an important characteristic that determines the service life of parts with PCMs in metal–polymer tribo compounds.

Density was measured by hydrostatic weighing on VLA-200-M scales to the nearest 2 mg. The accuracy of density measurement was up to 0.1%.

Tests for breaking strength and elongation at break were performed on ring specimens ø50 × ø40 and 10 mm high using hard semi-disks on the rupture unit MR-05-1 at a speed of 10 mm/min and a load of 100 kgf (980.655 N). The error of load measurement was not more than 1% of the measured value, and the definition of geometric dimensions was not more than 0.05 mm.

Breaking strength (σ_b_) (MPa) was calculated by the formula:σ_b_ = P/2hh_1_ = P/2S,(1)
where P—breaking force, H (kgf); h—radial wall thickness of the annular sample, m (cm); h_1_—axial height of the annular specimen, m (cm); S—minimum cross section of the annular sample, m (cm).

The method is based on the stretching of the test specimens with a set strain rate, at which the indicators of Formula (1) are determined. During testing, the load and elongation of the specimens are measured continuously or at the moment of reaching the yield strength, maximum load, or at the moment of destruction of the specimens, as in our case. Samples in which defects were found during the test (bubbles, inclusions foreign to the material, internal cracks, etc.) are not considered.

Relative elongation at break δ (%) was calculated by the formula:δ = Δl/l_0_ × 100%,(2)
where Δl—change in the calculated length of the sample at the time of rupture, mm; l_0_—initial estimated length of the sample, mm.

Investigations of wear intensity were performed on a serial friction machine 2070 SMT-1 according to the scheme “partial insert–shaft”. The counter body was a roller ø48 mm made of steel 45 (HRC 45, Ra 0.72 μm). The partial insert was made of the test material and was a section 16 mm wide with a ring ø80 by ø60 mm and a height of 9 mm.

Run-in and testing for each of the samples of material was performed on one track. The amount of wear of the samples was determined gravimetrically on analytical balances with an accuracy of 10^−5^ g and counted on the intensity of wear according to the formula:I = V/P·S,(3)
where V—volume of worn material, mm^3^; P—normal load, H; S—path of friction, m.

When estimating the intensity of PCM wear, the root mean square error was regulated by errors in measuring the mass of the sample, speed, and duration of friction and did not exceed 5%.

The study of the microstructure of PCMs was performed using a scanning electron microscope of high resolution, TESCAN MIRA 3 LMU.

## 3. Numerical Approach Used for Simulation

Based on the basic principles of building three-dimensional structural models of composites based on statistics on the strength of inclusions and matrices, a finite element model was built in ANSYS APDL to simulate the strength and load-bearing capacity of the material with analysis of damage accumulation in the model. In structural models, the mechanical interaction of the microstructures of the filler and the matrix is simulated. This approach is called structural or micromechanical and has been actively developed in recent years [61,62,63].

The difficulties that arise in the creation of a reliable structural model are significant:increased requirements for the accuracy of determining the stress–strain state of composite components, as the onset of composite fracture is usually associated with local physical processes, so it is impossible to use many simple structural models sufficient to analyze the integral (e.g., stiffness) characteristics of the composite.the need to consider the kinetics of fracture of the material because the local values of the parameters of the stress–strain state of the composite components often reach the limit values at the initial stages of loading the composite, but this does not lead to the depletion of its bearing capacity.

### 3.1. Construction of a Solid Model of a Composite Material

In order to numerically determine the elastic properties of the material chaotically reinforced with spherical and cylindrical inclusions, the composite material was simulated by the finite element method in a three-dimensional setting. The matrix of the composite material was industrial PTFE with elastic properties: modulus of longitudinal elasticity E = 410 MPa and Poisson’s ratio ν = 0.45. The filler of spherical inclusions was fine coke with a modulus of longitudinal elasticity E = 500 MPa and Poisson’s ratio ν = 0.30. The filler of cylindrical inclusions was a carbon fiber with a modulus of longitudinal elasticity E = 35,000 MPa and a Poisson’s ratio ν = 0.25. The geometric shape of the spherical filler was spherical, with diameters di from 10 to 50 μm. For the cylindrical filler, straight circular cylinders with diameters di from 10 to 12 μm and length of generating li from 100 to 150 μm were chosen.

The solid model adopted a cube with rib length *a* with the properties of the matrix, saturated with spherical or cylindrical inclusions with the properties of the filler (Figure 2). The number of filler bodies was considered in order to achieve its volumetric content in the composite of 20% for spherical and 15% for cylindrical.

#### 3.1.1. Modeling of Spherical Filler

The spherical filler was modeled by lottery with a uniform probability of the current diameter of the ball di in the range from 10 μm to 50 μm, respectively, and the position of the center (*x_i_; y_i_; z_i_*) in the range from *d_i_/2* μm to *a − d_i_/2* μm so that the bullet of the filler was guaranteed not to reach beyond the cube. This made it possible to exclude from consideration the issue of modeling the dissection of the filler, which is a separate problem of mechanical processing. The lottery was built in a solid model provided that it did not intersect with pre-built balls:(4)$$\sqrt{{\left(x_{i}-x_{j}\right)}^{2}+{\left(y_{i}-y_{j}\right)}^{2}+{\left(z_{i}-z_{j}\right)}^{2}}>\frac{d_{i}-d_{j}}{2},j=1\ldots (i-1)$$
where *i*—number of the current ball, *j*—number of the ball already accepted for construction.

#### 3.1.2. Modeling of Cylindrical Filler

The cylindrical filler was modeled by lottery with a uniform probability of the current diameter of a straight circular cylinder d_i_ in the range from 10 to 12 μm, respectively, the length of the generating l_i_ from 100 to 150 μm, the position of the center (*x_i_; y_i_; z_i_*) from *d_i_/2* μm to *a* − *d_i_/2* μm, so that the center of mass of the filler cylinder was guaranteed not to reach beyond the cube, and the cosine guides (*a_xi_; a_yi_; a_zi_*). Since d_i_ is an order of magnitude less than l_i_, we exclude the issue of modeling the cylinder end within the simulated volume, which significantly complicates the condition of non-intersection of the filler cylinders. Thus, in the model, the cylinders are forcibly crossed by the side surfaces of the cube face—the matrix model—which imposes an additional condition: the length of the truncated face of the cylinder cube is no longer than the maximum length of 150 μm. 

The lottery was built in a solid model provided that it did not intersect with pre-built balls. This condition, in view of the above, was simplified to the condition that the distance between the lines drawn through the center of mass of the cylinders (*x_i_; y_i_; z_i_*) in the directions (*a_xi_; a_yi_; a_zi_*) was greater than the sum of their diameters:(5)$$\frac{\left(x_{i}-x_{j}\right)\cdot \left(a_{y_{i}}\cdot a_{z_{j}}-a_{z_{i}}\cdot a_{y_{j}}\right)+\left(y_{i}-y_{j}\right)\cdot \left(a_{z_{i}}\cdot a_{x_{j}}-a_{x_{i}}\cdot a_{z_{j}}\right)}{\sqrt{{\left(a_{y_{i}}\cdot a_{z_{j}}-a_{z_{i}}\cdot a_{y_{j}}\right)}^{2}+{\left(a_{z_{i}}\cdot a_{x_{j}}-a_{x_{i}}\cdot a_{z_{j}}\right)}^{2}+{\left(a_{x_{i}}\cdot a_{y_{j}}-a_{y_{i}}\cdot a_{x_{j}}\right)}^{2}}}+\;\frac{\left(z_{i}-z_{j}\right)\cdot \left(a_{x_{i}}\cdot a_{y_{j}}-a_{y_{i}}\cdot a_{x_{j}}\right)}{\sqrt{{\left(a_{y_{i}}\cdot a_{z_{j}}-a_{z_{i}}\cdot a_{y_{j}}\right)}^{2}+{\left(a_{z_{i}}\cdot a_{x_{j}}-a_{x_{i}}\cdot a_{z_{j}}\right)}^{2}+{\left(a_{x_{i}}\cdot a_{y_{j}}-a_{y_{i}}\cdot a_{x_{j}}\right)}^{2}}}>\frac{d_{i}+d_{j}}{2},j=1\ldots (i-1)$$
where *i*—number of the current ball, *j*—number of the ball already accepted for construction. To smooth the effects of joining different modular materials, loading and fixing took place through additional upper and lower cubes with matrix properties.

### 3.2. Construction of a Finite Element Model of Composite Material

The solid model was divided into three-dimensional finite elements and symmetrical boundary conditions were applied over the area of the lower face (lower) of the cube; see Figure 3. All volumes of simulated matrices and fillers touched with perfect contact, i.e., the nodes of the elements on common surfaces had common displacements; see Figure 4. 

Additionally, one of the lower vertices (lower) of the cube was fixed from linear displacements (spherical hinge) to ensure the kinematic immutability of the model. The loading was applied by negative pressure on the upper face (upper) of the cube, which was equal to the tensile strength of the corresponding composite material. The tensile process of the sample with normal stress corresponding to the experimental material strength limit was modeled: 18.6 MPa for spherical filler and 19.1 MPa for cylindrical filler.

## 4. Results and Discussion

### 4.1. Experimental Results

#### 4.1.1. Influence of Coke Concentration on Properties of PTFE PCMs 

A study was performed on the effect of different concentrations of dispersed coke filler to determine its optimal content in mechanoactivated PTFE (Table 2). 

Its optimal concentration in the composite was 20 wt.%. At the same time, the minimum value of wear intensity at the necessary level of mechanical properties was reached.

The use of a mechanically activated matrix helps to increase the properties of PTFE PCMs with coke: breaking strength is increased by (2.8−7.5)%, and relative elongation at break is increased by (2.6−10)%, with a decrease in the intensity of wear by (5.6−18)% compared to composites based on inactivated PTFE. 

The addition of coke to the mechanically activated PTFE matrix, depending on the concentration of the latter, reduces its level of breaking strength by (27–44)% and relative elongation at break by (2.8–3.7) times while reducing the wear intensity by 9–15 times.

#### 4.1.2. Influence of CF Concentration on Properties of PTFE PCMs 

A study was performed on the effect of different concentrations of fibrous filler to determine its optimal content in mechanoactivated PTFE (Table 3). The optimal concentration of explosive fragments in the composite was 15 wt.%, which corresponds to the formation of a more homogeneous structure of the composite and high physical and mechanical and tribotechnical properties.

The use of a mechanically activated matrix helps to increase the properties of PTFE PCMs filled with CF: breaking strength is increased by (2.2−8.9)%, and relative elongation at break is increased by (8.7−20.8)%, with a decrease in the intensity of wear by 18% compared with composites based on inactivated PTFE. 

The addition to the mechanically activated PTFE matrix of CF, depending on the concentration of the latter, reduces its level of breaking strength by (12−39)% and its relative elongation at break by (2.9−4.2) times, with a significant reduction in wear intensity by (17−25) times.

#### 4.1.3. Influence of Mechanical Activation of PTFE Matrix and Fillers on Properties of PCMs 

The most defective place in the structure of filled polymer systems is the boundary layer between the matrix and the filler particles, because the destruction of the material usually occurs at the interfacial boundaries. Therefore, in order to obtain the required level of adhesive interaction at the interface of the filled composite, it is necessary that both the polymer macromolecule and the filler particle have enough surface activity. To achieve this, we used mechanical activation to prepare the ingredients of the composition to obtain PCMs. 

The synergistic effect of the application of mechanical activation of both matrix PTFE and fillers before mixing was manifested in increased performance of the obtained composites, which indirectly indicates an increase in the adhesive polymer–filler interaction (Table 4).

Analysis of Table 4 shows that the mechanical activation of both the matrix and the filler contributes to a significant increase in PTFE composites: when filling 15 wt.% CF, breaking strength increases by 9.5% and 18.6%, relative elongation at break is increased by 6.2% and 28%, and wear intensity reduces by 5.4 times and 6.4 times compared to inactivated CF and inactivated PTFE, respectively;when filling 20 wt.% coke, breaking strength increases by 8.1% and 16.3%, relative elongation at break is increased by 4.5% and 15%, and wear intensity reduces by 1.4 times and 1.7 times compared to inactivated coke and inactivated PTFE, respectively

### 4.2. Model Simulation Results 

Equivalent von Mises stress values for matrices and fillers are shown in Figure 5, Figure 6 and Figure 7.

### 4.3. Discussion

In general, it can be argued that the finite element model predicts well the strength of chaotically reinforced spherical inclusions of composite materials: the matrix and the filler reach the strength limit at the same time. The polytetrafluoroethylene matrix in tensile fracture has equivalent Mises stresses of 21.5 MPa against 23 MPa in tabular data and spherical coke filler has a value of 23.4 MPa against 19.6 MPa in tabular data, which corresponds to the experimental data. This is due to the smoothness of the filler surfaces and the lack of filler dissection in the model.

The prediction of the strength of a finite element model of chaotically reinforced cylindrical inclusions of composite materials requires additional analysis. The matrix and the filler obviously have stress concentrators and are both subject to the difficulties of creating a reliable structural model. Analysis of Figure 6 and Figure 7 allows us to conclude that on the line between the filler and the matrix of the dissected surface of the filler, there are stress concentrators, which should lead to the extraction of fibers oriented in space approximately in the load direction. Similarly, on the line of the boundary of the matrix with the filler near the dissected surface of the filler, the matrix should be chipped. However, by removing the filler hub region from consideration and allowing the adhesion of the matrix to the filler to be destroyed, which will significantly unload the matrix, the strength of the composite material can be predicted.

In addition, the presence of dissection of the filler in the simulation leads to the following consequences:From the point of view of the accuracy of determining the stress–strain state, it is necessary to consider the boundary conditions applied to the cut surface of the filler:
under the boundary conditions formulated in stresses, anomalous displacement in proportion to the stiffness will be inappropriate for the experiment;when formulating the kinematic boundary conditions, the stress state will behave as a classical stress concentrator—local stress perturbation.From the point of view of accuracy in determining the kinetics of material destruction, common nodes of elements on the boundary line with the matrix of the dissected surface of the filler may have mixed boundary conditions, which can lead to errors in numerical solution (artifacts).

Therefore, in further research, it would be rational to use the effects of cohesion due to the appropriate settings of the contact between the filler and the matrix or introduction of special interface elements into the gap between them using Cohesive Zone Modeling technology, which models interface delamination and progressive failure where two materials are joined together.

## 5. Conclusions

The results of the experimental study of the effect of the concentration of dispersed and fibrous filler on the physical and mechanical properties of PTFE PCMs allowed us to determine the optimal content of filler in the volume of the polymer matrix: 20 wt.% for coke and 15 wt.% for carbon fiber. 

Using the mechanical activation of both the matrix and the fillers before its mixing contributes to a significant increase in PTFE PCMs: when filling 15 wt.% CF, breaking strength increases by 9.5% and 18.6%, relative elongation at break increases by 6.2% and 28%, and wear intensity reduces by 5.4 times and 6.4 times compared to inactivated CF and inactivated PTFE, respectively; when filling 20 wt.% coke, breaking strength increases by 8.1% and 16.3%, relative elongation at break increases by 4.5% and 15%, and wear intensity reduces by 1.4 times and 1.7 times compared to inactivated coke and inactivated PTFE, respectively.

For numerically determining the strength of materials chaotically reinforced with spherical and cylindrical inclusions, the composite material was simulated by the finite element method in a three-dimensional setting in ANSYS APDL. The simulation results showed that the finite element model predicts well the strength of chaotically reinforced spherical inclusions of composite materials. The PTFE matrix under tensile fracture has equivalent Mises stresses of 21.5 MPa against 23 MPa in tabular data and spherical coke filler has a value of 23.4 MPa against 19.6 MPa in tabular data, which corresponds to the experimental data. 

The prediction of the strength of a finite element model of chaotically reinforced cylindrical inclusions of composite materials requires additional analysis. On the line of the fiber boundary with the matrix of the dissected surface of the filler, there are stress concentrators, which should lead to the extraction of the fibers, oriented in space approximately in the direction of the load. Similarly, on the line of the boundary of the matrix with the filler near the dissected surface of the filler, the matrix should be chipped. The model does not consider these features. However, by removing the filler hub region from consideration and allowing the adhesion of the matrix to the filler to be destroyed, which will significantly unload the matrix, the strength of the composite material can be predicted.

In general, the results of this study and the obtained finite element models for predicting the strength of reinforced PTFE with a chaotic arrangement in the volume of spherical and cylindrical inclusions can be applied to the matrices of other thermoplastic polymers but given their structural features. Moreover, instead of the CF and coke studied in this case, researchers can use different cylindrical and spherical fillers to create their finite element model for predicting the strength of PCMs.

## Figures and Tables

**Figure 1 polymers-14-02087-f001:**
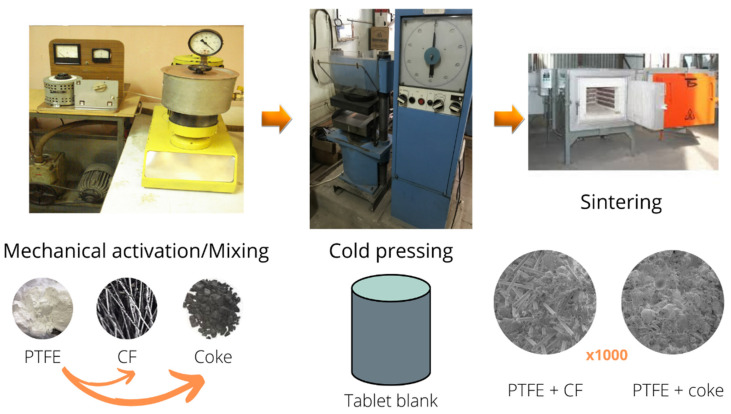
Scheme of step-by-step production technology of PTFE PCMs.

**Figure 2 polymers-14-02087-f002:**
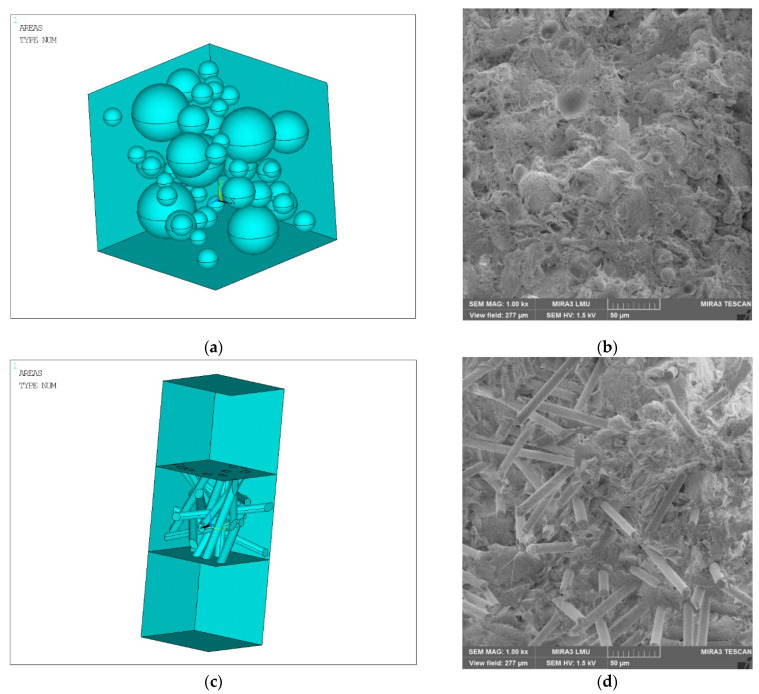
Solid model of composite materials, chaotically reinforced with spherical and cylindrical inclusions (**a**,**c**) (all areas of the simulated volumes are shown, except the ones closest to the observer) and real microstructures of PTFE composite with 20% coke (**b**) and 15% CF (**d**).

**Figure 3 polymers-14-02087-f003:**
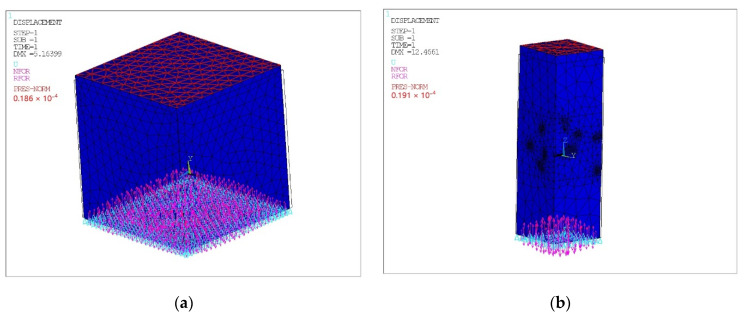
Deformable state of a finite element model of composite materials chaotically reinforced with spherical (**a**) and cylindrical (**b**) inclusions with applied boundary conditions and reactions.

**Figure 4 polymers-14-02087-f004:**
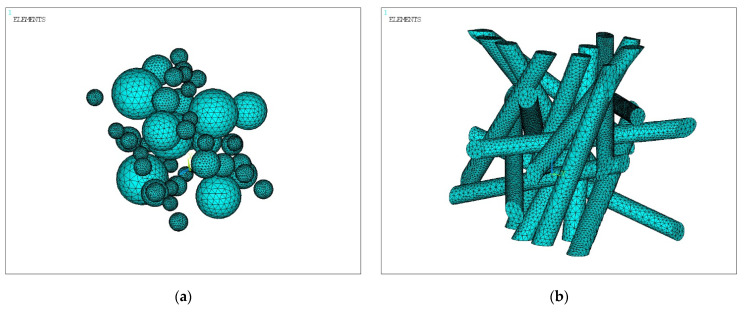
Finite element models of fillers with spherical (**a**) and cylindrical (**b**) inclusions.

**Figure 5 polymers-14-02087-f005:**
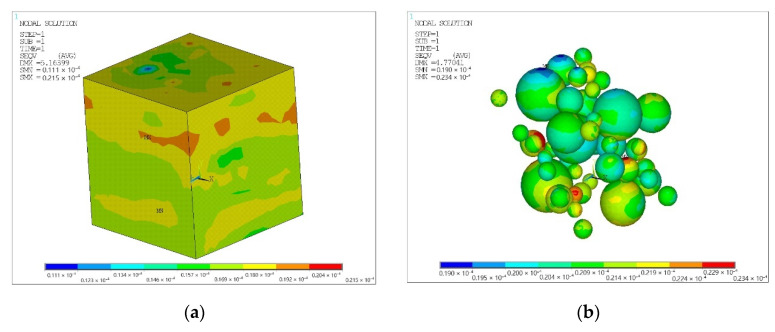
Equivalent von Mises stress for matrix (**a**) with spherical filler (**b**), (10^−6^ MPa).

**Figure 6 polymers-14-02087-f006:**
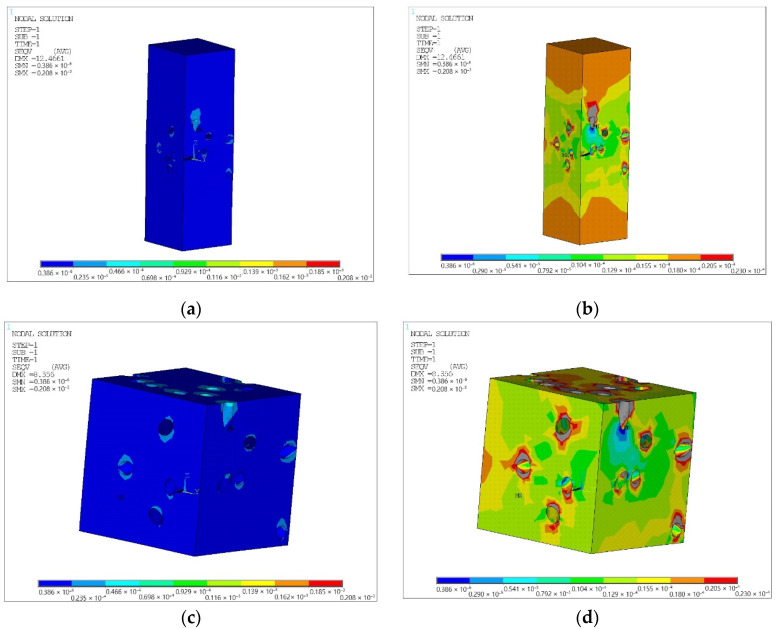
Equivalent von Mises stress for a matrix with a cylindrical filler, (10^−6^ MPa): (**a**) for the complete calculation model without restrictions on stress values; (**b**) for the complete calculation model with stress values up to the matrix strength limit of 20.8 MPa; (**c**) for the part of the calculation model that is saturated with fibers, without restrictions on the values of stresses; (**d**) for the part of the calculation model saturated with fibers, with stress values up to the matrix strength limit of 20.8 MPa.

**Figure 7 polymers-14-02087-f007:**
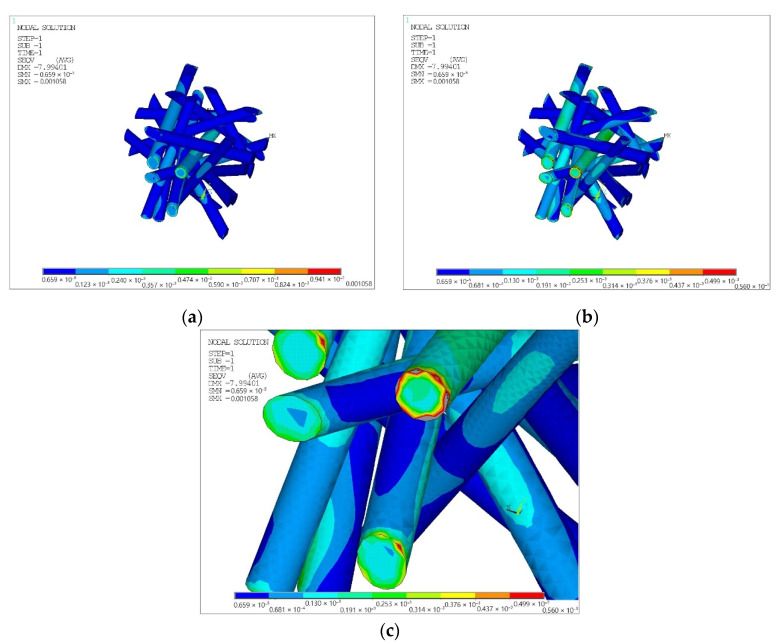
Equivalent von Mises stress for cylindrical filler, (10^−6^ MPa): (**a**) without restrictions on the magnitude of stresses; (**b**) with stress values up to the strength limit of the cylindrical filler 560 MPa; (**c**) with stress values up to the tensile strength of the cylindrical filler 560 MPa in the area of excess (the view from the opposite side is similar and not shown).

**Table 1 polymers-14-02087-t001:** Physical and mechanical properties of materials for research.

Parameter	PTFE	CF	Coke
Size, μm	50–500	d = 10–12; l = 100–150	10–50
Density, kg/m^3^	2200	1510	1730
Tensile strength, MPa	23	520–600	15–25
Compressive strength, MPa	11.8	500–550	9.8–19.6
Modulus of elasticity, MPa	410	27–47	500
Poisson’s ratio	685.5	0.10–0.30	0.30

**Table 2 polymers-14-02087-t002:** The effect of coke concentration on the properties of PTFE composites.

Concentration of Coke (wt.%)	Density ρ, kg/m^3^	Breaking Strength σ_b_, MPa	Relative Elongation at Break δ, %	Wear Intensity I × 10^−6^, mm^3^/N·m
95:5	2130/2145 *	18.9/19.5 *	145/150 *	67.5/63.0 *
90:10	2115/2125 *	17.6/18.1 *	132/142 *	63.5/60.0 *
85:15	2105/2115 *	16.9/17.7 *	115/118 *	56.5/53.5 *
80:20	2090/2100 *	16.0/17.2 *	100/110 *	47.5/40.0 *

Note: *—in the numerator data for the inactivated matrix, and in the denominator—after mechanical activation.

**Table 3 polymers-14-02087-t003:** The effect of CF concentration on the properties of PTFE composites.

Concentration of CF (wt.%)	Density ρ, kg/m^3^	Breaking Strength σ_b_, MPa	Relative Elongation at Break δ, %	Wear Intensity I × 10^−6^, mm^3^/N·m
90:10	2010/2020 *	17.5/17.9 *	90/98 *	42.5/36.0 *
85:15	1980/1990 *	20.4/22.1 *	120/145 *	32.0/27.0 *
80:20	1960/1980 *	18.3/19.1 *	105/115 *	35.0/29.5 *
75:25	1950/1960 *	16.9/18.4 *	115/125 *	29.0/24.5 *

Note: *—in the numerator data for the inactivated matrix, and in the denominator—after mechanical activation.

**Table 4 polymers-14-02087-t004:** Properties of composites based on activated PTFE and fillers.

Composition of PCM (wt.%)	Density ρ, kg/m^3^	Breaking Strength σ_b_, MPa	Relative Elongation at Break δ, %	Wear Intensity I × 10^−6^, mm^3^/N·m
80 PTFE:15 CF	1980/1990 *	22.1/24.2 *	145/154 *	27.0/5.0 *
80 PTFE:20 coke	2100/2110 *	17.2/18.6 *	110/115 *	40.0/28.0 *

Note: *—in the numerator data for the inactivated fillers, and in the denominator—after mechanical activation.

## Data Availability

The data presented in this study are available on request from the corresponding author.

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
