# Peer review of "Modeling of Polymer Composite Materials Chaotically Reinforced with Spherical and Cylindrical Inclusions"

_polymers, 2022, doi:10.3390/polym14102087_

Round 1

Reviewer 1 Report

The article is devoted to an actual problem, the relevance of which is confirmed by a wide literature review. The article clearly outlines the purpose and objectives that were solved in this article. A large amount of research has been carried out. Interesting results have been obtained. The article is written in fine English and is recommended for publication. To improve the quality of the article, the following minor corrections are recommended:

– Line 138. Please decipher the abbreviation of CF;

– Line 142. Check the correctness of the phrase “The fine coke foundry coke…”;

– Tables 2-4 use the term "Wear intensity". However, in the text, when describing these Tables, the term "wear resistance" is used. Probably, the authors have in mind the same parameter. It is recommended to use either of the two terms;

– Section 5 "Discission" is best combined with section 4 "Results". It is also recommended to change the name of section 4 "Results" to "Results and Discussions".

Author Response

The article is devoted to an actual problem, the relevance of which is confirmed by a wide literature review. The article clearly outlines the purpose and objectives that were solved in this article. A large amount of research has been carried out. Interesting results have been obtained. The article is written in fine English and is recommended for publication. To improve the quality of the article, the following minor corrections are recommended:

– Line 138. Please decipher the abbreviation of CF;

We have added a decoding of the abbreviation.

– Line 142. Check the correctness of the phrase “The fine coke foundry coke…”;

Thank you for your valuable comment, there was an inaccuracy in the translation. It was fixed.

– Tables 2-4 use the term "Wear intensity". However, in the text, when describing these Tables, the term "wear resistance" is used. Probably, the authors have in mind the same parameter. It is recommended to use either of the two terms;

Wear resistance is the property of a material to resist wear under certain friction conditions, estimated by the reciprocal of wear intensity. Therefore, these two concepts are not the same, but have opposite meanings.

We listened to the reviewer's remark and, in order not to mislead readers, we leave only one designation "wear intensity", changing the value in the explanations of the tables accordingly.

– Section 5 "Discission" is best то combined with section 4 "Results". It is also recommended to change the name of section 4 "Results" to "Results and Discussions".

We listened to the reviewer's remark and combined sections "Results" and "Discussion" in one "Results and Discussions".

Reviewer 2 Report

Comments

This paper studied Modeling of Polymer Composite Materials. The outcome of the paper is interesting however, there are several aspects that need to be improved. The reviewer can only recommend for publication if the author satisfactorily address the following major comments in the revised version.

  1. The research gap from the literature review should be clearly presented.
  2. The research questions and justification of selecting variables should be highlighted.
  3. The failure mechanism of the specimen should be discussed more clearly.
  4. The novelty of the study should be highlighted more clearly at the end of introduction section. How this study is different from the published study in literature?
  5. How the outcome of this study will benefit researchers and end users? This need to be highlighted in introduction or end of conclusion.
  6. The fillers in composite materials is interesting but not novel. Therefore, the recent application in this area should be discussed in introduction section to improve the background study. Recently, fillers was applied for polymer concrete [Ref: Investigation on the physical, mechanical and microstructural properties of epoxy polymer matrix with crumb rubber and short fibres for composite railway sleepers], and geopolymer concrete [Ref: Effect of short fibres in the mechanical properties of geopolymer mortar containing oil-Contaminated sand]. Suggest to include them in introduction section with proper citations to improve the background study.

I would be happy to see the revised version to understand how these comments are being addressed.

Author Response

Dear Reviewer,  thank you for the review.

The research gap from the literature review should be clearly presented.

It was added.

The research questions and justification of selecting variables should be highlighted.

It was added.

The failure mechanism of the specimen should be discussed more clearly.

Clarification of the conditions for the tensile test has been added.

The novelty of the study should be highlighted more clearly at the end of introduction section. How this study is different from the published study in literature?

It was added.

How the outcome of this study will benefit researchers and end users? This need to be highlighted in introduction or end of conclusion.

It was added.

The fillers in composite materials is interesting but not novel. Therefore, the recent application in this area should be discussed in introduction section to improve the background study. Recently, fillers was applied for polymer concrete [Ref: Investigation on the physical, mechanical and microstructural properties of epoxy polymer matrix with crumb rubber and short fibres for composite railway sleepers], and geopolymer concrete [Ref: Effect of short fibres in the mechanical properties of geopolymer mortar containing oil-Contaminated sand]. Suggest to include them in introduction section with proper citations to improve the background study.

The proposed papers were included and discussed in introduction section for improving the background study.

Round 2

Reviewer 2 Report

I have no further comments